# Structural basis for antibody cross-neutralization of Dengue and Zika viruses
Nicholas K. Hurlburt [1], Jay Lubow [1], Leslie Goo[1,2] & Marie Pancera [1] ✉

Safe and effective vaccines against co-circulating mosquito-borne orthoflaviviruses such as Zika virus (ZikV) and the four serotypes of Dengue virus (DenV1-4) must elicit broadly neutralizing antibodies (bnAbs) to prevent the risk of enhancement of infection by non-neutralizing antibodies. We recently discovered new orthoflavivirus-directed bnAbs, including F25.S02, which neutralizes DenV1-4 and ZikV with comparable or superior potency to the previously characterized E dimer epitope (EDE) bnAbs. Here, we used cryoEM and X-ray crystallography to understand the basis of cross-neutralization of F25.S02 at the molecular level. We obtained a ~ 4.2 Å cryoEM structure of F25.S02 Fab bound to a stabilized DenV3 soluble E protein dimer and a 2.3 Å crystal structure of F25.S02 Fab bound to ZikV soluble E protein dimer. Like previously described EDE1 bnAbs, the structural epitope of F25.S02 is at the E dimer interface, encompassing predominantly conserved regions in domain II, including the fusion loop. However, unlike EDE1 bnAbs, F25.S02 binding is almost entirely dependent on the heavy chain and is shifted slightly away from the dimer symmetry axis. Our findings emphasize the importance of this cross-neutralizing site of vulnerability for DenV and ZikV that can facilitate rational design of vaccines and therapeutics.

Orthoflaviviruses are arthropod-borne, enveloped RNA viruses that can cause symptoms ranging from a self-limiting fever to potentially fatal hemorrhagic or neuroinvasive disease for which there is no treatment beyond supportive care. Many orthoflaviviruses are emerging threats to global health: the four DenV serotypes (DenV1-4) infect an estimated 390 million people annually[1]; ZikV emerged outside of Asia and Africa for the first time in 2007 and was repeatedly introduced into the Americas in 2014–2015, leading to considerable morbidity[2]. The ability of ortho-flaviviruses to rapidly emerge from relative obscurity to serious public health threats combined with the geographical expansion of mosquito vectors[3], further facilitated by climate change[4], underscores the importance of effective clinical interventions.

The envelope (E) protein of orthoflaviviruses, including DenV1-4 and ZikV, is a key structural protein involved in viral entry and membrane fusion, and is the target of antibodies[5]. The soluble form of this protein (sE) retains many of the antigenic properties of the native virion-associated E protein[6]. While structurally conserved across DenV1–4 and ZikV, variations in sE can influence virus-specific interactions with host antibodies and receptors, impacting cross-reactivity and immune protection[7].

The complex antibody response to co-circulating and antigenically related orthoflaviviruses presents a major challenge for vaccine development. Specifically, the presence of cross-reactive, non-neutralizing antibodies to E protein from a prior exposure to ZikV or a given DenV serotype predicts the risk of severe disease following secondary exposure to a different DenV serotype[8,9] - a process known as antibody-dependent enhancement (ADE) of infection. To minimize the risk of ADE, a safe and effective vaccine must elicit antibodies that simultaneously neutralize DenV1-4, and ideally, ZikV. Despite decades of research, ADE-related safety concerns ultimately derailed the widespread use of the first licensed DenV vaccine[10]. This unfortunate outcome eroded public confidence in vaccines in general[11]. Thus, although many successful vaccines have been developed without first knowing the specific targets of the antibodies they elicited, such empirical approaches to developing DenV vaccines have been insufficient. A rational approach to vaccine design to limit the elicitation of non-neutralizing antibodies is needed and this requires improving our fundamental understanding of how orthoflavivirus bnAbs recognize their targets.

A few bnAbs that can potently neutralize DenV1-4 and in some cases, ZikV, have been isolated from infected individuals. The most well-characterized bnAbs target an envelope-dimer epitope (EDE) spanning both E protein protomers within the dimer subunit as defined by X-ray crystallography and cryo-electron microscopy (cryoEM)[12,13]. There are two subclasses of EDE bnAbs, of which EDE1 but not EDE2 can potently neutralize ZikV in addition to DenV1-4[14]. Until recently, SIgN-3C was the

[1]Vaccine and Infectious Disease Division, Fred Hutchinson Cancer Center, Seattle, Washington, USA. [2]Present address: Vaccine Company, Inc., South San Francisco, San Mateo, CA, USA. ✉e-mail: mpancera@fredhutch.org

only known naturally occurring, cross-neutralizing antibody outside the EDE1 class[15]. Unlike EDE antibodies, which target an *intra*-dimer epitope, SIgN-3C binds an epitope at the *inter*-dimer interface[16], as determined by cryoEM.

Recently, we identified F25.S02, another naturally occurring bnAb with broad and potent cross-neutralization of DenV1-4 and ZikV[17]. Mutational and neutralization analysis showed that the F25.S02 epitope was distinct from the EDE1 bnAb C10[17]. Here, we determined the binding mechanism of F25.S02 to DenV3 sE dimer by cryoEM and to ZikV sE dimer using X-ray crystallography. The structures revealed that F25.S02 differs in its mode of binding compared to other cross-neutralizing EDE1 antibodies. Our results can ultimately guide structure-based vaccine design, the promise of which is best exemplified by the recently approved vaccine for respiratory syncytial virus[18].

## Results

### F25.S02 binds a highly conserved patch on DenV3 sE

We previously showed that F25.S02 had the highest neutralization potency against DenV3[17]. We expressed and purified a previously described stabilized sE protein dimer (DenV3 sE sc30)[19] and a His-tagged antigen binding fragment (Fab) of F25.S02. We formed a stable complex as indicated by a monodisperse peak by size-exclusion chromatography (SEC) and confirmed by SDS-PAGE (Supplementary Fig. 1a, b). This complex was used to obtain a cryoEM reconstruction to 4.2 Å resolution (Fig. 1a, Supplementary Fig. 2, Table 1). The map revealed the sE dimer with two Fabs bound at the envelope dimer interface, similar to what has been shown with previously identified EDE1 antibodies[14,20]. F25.S02 interacts with domain I and III of one protomer and domain II and fusion loop of the other protomer (Fig. 1a). The local resolution of the F25.S02 for most of the binding site is around 3.5 Å, increasing the confidence in sidechain assignment (Supplementary Fig. 2d, e). F25.S02 binds DenV3 with a total buried surface area (BSA) of ~1157 Å[2] with ~1010 Å[2] from the heavy chain (HC) and ~147 Å[2] from the light chain (LC). Thus, nearly 87% of the BSA comes from the HC (Fig. 1b, Supplementary Fig 3, Supplementary Table 1). The F25.S02 binding site is a highly conserved patch on the E protein among all DenV serotypes and

ZikV (Fig. 1c) as shown by mapping the surface with a conservation score generated by the Consurf server[21] using E protein sequences of reference strains of each DenV serotype and ZikV (DenV1: WP-74, DenV2: 16681, DenV3: CH53489, DenV4: TVP376, ZikV: H/PF/2013).

The cryoEM map shows clear density for the first N-acetylglucosamine (GlcNAc) of the N67 glycan (Supplementary Fig. 4a) and two GlcNAc and one mannose of the N153 glycan (Supplementary Fig. 4b); both glycans are conserved among DenV serotypes. F25.S02 does not contact the glycan at N67 but interacts with the glycan at N153. This was unexpected because we previously showed that mutations at either N153 or T155 to remove the glycan increased F25.S02 neutralization potency against DenV2[17]. To determine if the glycan was needed for DenV3 recognition, we removed the glycan by mutating threonine 155 to alanine (T155A) and tested binding of F25.S02 to DenV3 sE sc30 T155A by biolayer interferometry (BLI). The BLI data showed that the T155A mutation had no effect on F25.S02 Fab binding to DenV3 sE sc30, indicating that the N153 glycan is not necessary for binding (Supplementary Fig. 1c) even though it contributes ~213 Å[2] of buried surface area (18% of the total BSA for DenV3).

### F25.S02 binds ZikV sE at the E dimer interface

As F25.S02 cross-neutralized DenV and ZikV, we next determined its structure bound to ZikV to understand the basis of its cross-reactivity. The wildtype (WT) ZikV sE dimer showed high affinity binding for F25.S02 (Supplementary Fig. 1d). To increase yield for structural studies, we co-expressed ZikV sE WT with F25.S02 Fab and obtained enough complex for crystallization experiments. The purified complex was confirmed by SEC and SDS-PAGE to be sE dimer with two Fabs bound (Supplementary Fig. 1e, f). Well diffracting crystals were obtained, and an X-ray structure was determined to 2.3 Å resolution (Fig. 2a, Table 2). The crystal had a $P4_12_12$ space group with an sE dimer and two Fabs in the asymmetric unit (ASU). The structure was solved by molecular replacement with $R_{work}$ and $R_{free}$ of 0.225 and 0.274, respectively (Table 2). Like in the DenV3 structure, the ZikV structure shows that the F25.S02 epitope is mostly HC dependent, accounting for ~81% of the total BSA (Fig. 2b, f). The total BSA of F25.S02 on ZikV is ~1361 Å[2] with ~1104 Å[2] from the HC and ~257 Å[2] from the LC

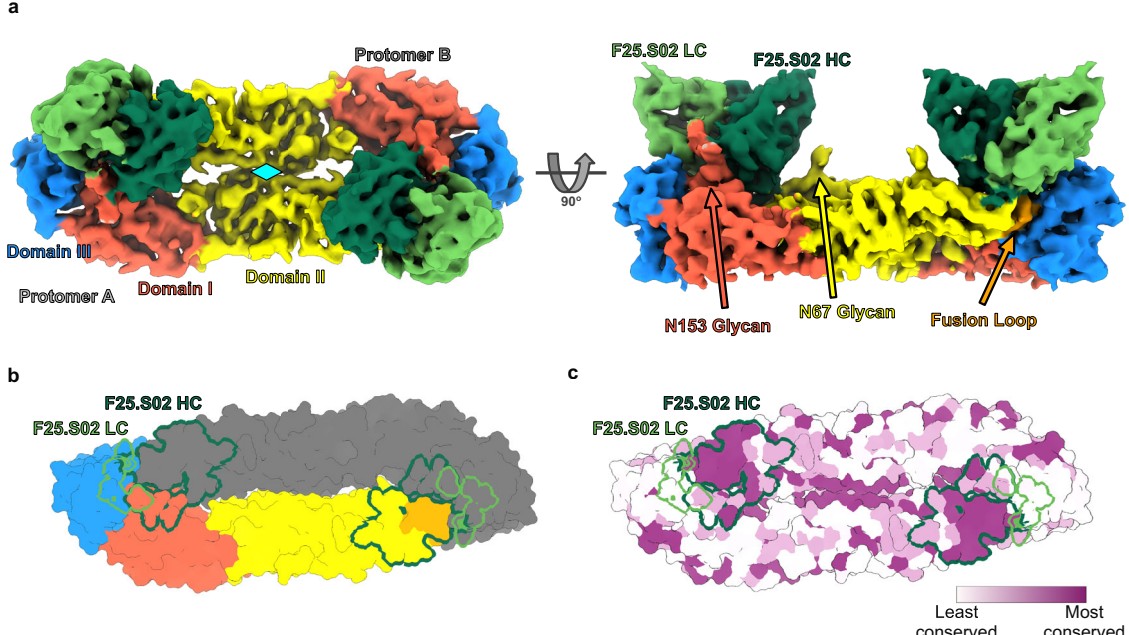

**Fig. 1 | CryoEM structure of F25.S02 bound to DenV3 sE. a** The cryoEM map of F25.S02 Fab bound to DenV3 sE sc30. The map is colored to highlight the domains of the sE protein with the glycans labeled and their locations shown. F25.S02 Fv is shown in dark green for the heavy chain and light green for the light chain. Cyan diamond represents the 2-fold symmetry axis of the dimer. **b** Model of DenV3 sE

sc30 dimer shown in surface representation with one protomer colored to show the domains and the other in gray. The binding footprint of F25.S02 is highlighted with the heavy chain binding site in dark green and the light chain in light green. **c** The same model from b) but colored to show conservation of residues across DenV serotypes and ZikV sE proteins. Coloring was generated using the ConSurf server.

**Table 1 | CryoEM data collection, refinement and validation statistics**

| | DenV3 sE sc30 bound to F25.S02 Fab (EMDB-70931) (pdb_00009owe) |
|---|---|
| **Data collection and processing** | |
| Magnification | 36,000 |
| Voltage (kV) | 200 |
| Electron exposure (e–/Å²) | 50 |
| Defocus range (μm) | −1.2 to −2.5 |
| Pixel size (Å) | 1.122 |
| Symmetry imposed | C2 |
| Initial particle images (no.) | 2,000,000 |
| Final particle images (no.) | 64,000 |
| Map resolution (Å) | 4.16 |
| FSC threshold | 0.143 |
| **Refinement** | |
| Initial model used (PDB code) | pdb_00007a3p |
| Model resolution (Å) | 4.16 |
| FSC threshold | 0.143 |
| Map sharpening $B$ factor (Å²) | −215 |
| Model composition | |
| Non-hydrogen atoms | 9710 |
| Protein residues | 1260 |
| Ligands | 104 |
| $B$ factors (Å²) | |
| Protein | 137.3 |
| Ligand | 130.4 |
| R.m.s. deviations | |
| Bond lengths (Å) | 0.002 |
| Bond angles (°) | 0.561 |
| Validation | |
| MolProbity score | 1.75 |
| Clashscore | 5.03 |
| Poor rotamers (%) | 0.95 |
| Ramachandran plot | |
| Favored (%) | 92.16 |
| Allowed (%) | 7.84 |
| Disallowed (%) | 0.00 |

(Supplementary Table 1). All three CDRHs are involved in binding but only the CDRL2 is involved in binding with single residues in framework region (FR) 2 and FR3 having minor contributions (Fig. 2f, Supplementary Fig. 3). When aligning the two Fabs the RMSD is 1.03 Å over 394 Cα with the biggest differences coming from the CH1/CL domain and when aligning just the residues involved in binding the RMSD is 0.218 Å over 23 Cα, indicating that the binding sites are in good agreement.

The majority of the interactions between F25.S02 and ZikV sE are located on domain II and the fusion loop of protomer A (Fig. 2c, Supplementary Fig. 4). The CDRH2 sits in a shallow pocket formed by the fusion loop, the β-strand *b* and the *ij* loop of domain II. The CDRH3 contains a one turn α-helix and sits on the surface formed by β-strand *b* and the fusion loop. Glu$_{73HC}$ in FR3 forms a H-bond with Arg$_{252ZikV}$ in domain II while Ser$_{74HC}$ of FR3 interacts with Asp$_{278ZikV}$ of protomer B. The CDRH1 of F25.S02 interacts with the 150 loop of domain I of protomer B, which contains the N154 glycan (Fig. 2d), that is equivalent to N153 in DenV (of note, ZikV

lacks the glycan at position 67 present in DenV). Interactions between F25.S02 LC and ZikV sE are all located on domain III with the CDRL2 forming three H-bonds with domain III of protomer B. The HC forms a single interaction with domain III with Leu$_{100EHC}$ sitting between Lys$_{316}$ of one protomer and the fusion loop of the other (Fig. 2e).

The ZikV N154 glycan is well resolved in the electron density, unlike in many other structures of antibodies bound to orthoflavivirus sE proteins[14,20]. For both protomers, electron density can clearly be seen up to the branching mannose, with electron density resolving the α1-3 and α1-6 mannose in protomer A while only the α1-3 mannose is seen with protomer B (Fig. 2d, Supplementary Fig. 4c, d). Interestingly, there are numerous interactions and H-bonds between F25.S02 and the N154 glycan. The first two N-terminal residues of the HC interact with the first and second GlcNAc and Arg$_{94HC}$ forms a hydrogen bond with the C3 hydroxyl of the first GlcNAc. Additionally, Lys$_{42LC}$ and Lys$_{45LC}$ interact with α1-3 mannose with Lys$_{45LC}$ forming a H-bond with the C4 hydroxyl (Fig. 2d). In total, the interactions with the N154 glycan account for ~20% of the total BSA (267 Å²) between F25.S02 and ZikV sE (Fig. 2f). Due to poor expression levels of ZikV sE WT, we were not able to produce enough to confirm if the removal of the glycan would affect F25.S02 binding affinity but it was previously shown that removal of the N154 glycan increased the neutralization potency of F25.S02 against ZikV[22], indicating that the glycan is not required for binding or neutralization.

**F25.S02 binds ZikV sE in a nearly identical manner as DenV3**

The residues of F25.S02 that interact with ZikV sE and DenV3 sE dimers are highly similar and most V-gene interactions are germline encoded, with only Met$_{51HC}$ being mutated from Ile (Fig. 2f). An alignment of the sE dimer of the ZikV and DenV3-F25.S02 bound structures shows good overlap with a root mean square deviation (RMSD) of 3.94 Å over 775 paired α carbons (Cα) (Fig. 3a). DenV3-bound F25.S02 Fv is shifted outward slightly relative to the ZikV-bound F25.S02 Fv, but this is mostly due to the slight difference in conformations of the sE proteins. Aligning domains I and III of protomer A and domain II of protomer B of DenV3 sE to the same regions of ZikV sE shows a much more similar structure with an RMSD of 1.36 Å over 320 paired Cα. An even more localized alignment using all residues of the sE proteins within 5 Å of F25.S02 had an RMSD of 0.99 Å over 36 Cα. When aligning the Fv domains of one of the F25.S02 Fabs in each structure, the overall RMSD is 1.02 Å over 234 paired Cα (Fig. 3b). In this alignment, the fusion loop and bound portions of domain II almost completely overlap. If you remove the interactions with the N153/N154 glycan from the BSA data, F25.S02 binds DenV3 sE with 85% HC and ZikV sE with 82% HC. The contact with the glycan in ZikV is ~209 Å² from the HC and ~58 Å² from the LC, while in DenV3, the HC and LC contact with an area of ~201 and ~13 Å², respectively (Supplementary Table 1, 2). The LC binding contribution to the glycans in the ZikV bound structure explains the difference between the 87% vs 81% overall BSA contribution of the HC to Denv3 versus ZikV. In all, F25.S02 primarily utilizes its HC to bind to the highly conserved fusion loop of DenV1-4 and ZikV. The light chain interactions are primarily with main chain atoms of ZikV sE that are not highly conserved, the only side chain H-bond is from Arg$_{54LC}$ to Glu$_{370ZikV}$ and a similar residue (Glu$_{370}$ in DenV1 and 3, Asp$_{370}$ in DenV2 and Asn$_{370}$ in DenV4) is present in DenV1-4 that would be capable of forming this interaction (Fig. 3c).

**F25.S02 binds a similar but distinct epitope to EDE1 antibodies**

We compared the binding mechanism of F25.S02 to two EDE1 bnAbs, C8[14] and C10[20], which also cross-neutralize DenV1-4 and ZikV. An alignment of the ZikV-bound C8 and C10 to ZikV-bound F25.S02 structure shows that both C8 and C10 have a larger footprint on domain II and bind closer to the dimer symmetry axis (Fig. 4a). Both C8 and C10 were shown to interact with N67 glycan on DenV (absent in ZikV) that is not required for binding or neutralization[14,20]. F25.S02 does not interact with this glycan on DenV3 (Fig. 3b). In the structures for C8 and C10, the 150-glycan loop is displaced by the CDRH3, and the glycan is not resolved. While in the

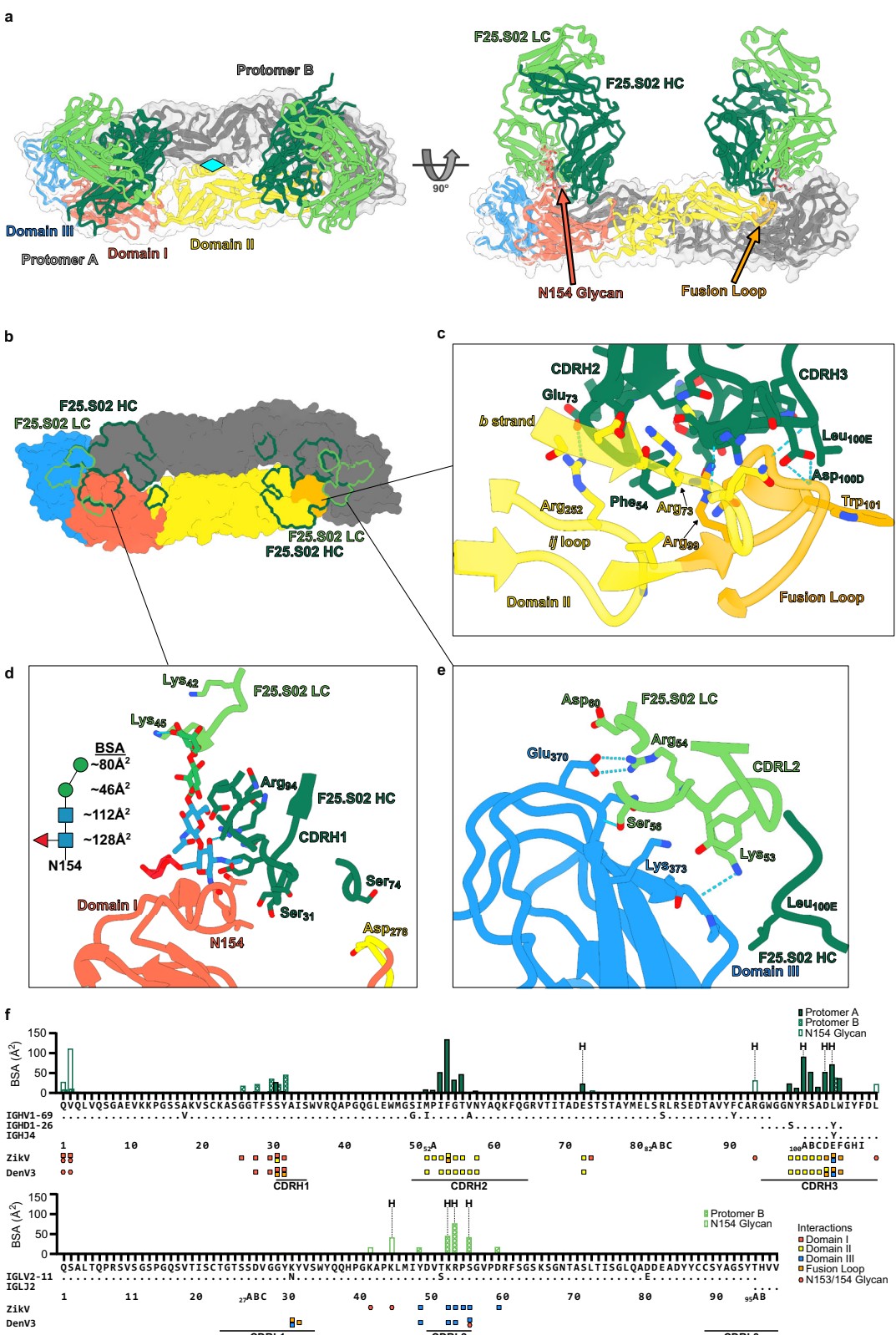

F25.S02 structures, the glycan is well-resolved and the CDRH1 interacts with the 150-loop staying in a capping position and the CDRH3 shifting to the top of the fusion loop relative to C8 and C10 further suggesting a different mode of binding. For both C8 and C10, the CDRL3 occupies the position of the F25.S02 CDRH3 (Fig. 4b), highlighting this shift towards the center of the sE dimer. With the binding sites aligned, a plane drawn

through the center of mass of each Fv domain further shows the difference in angle of approach (Supplementary Fig. 5a, b). An alignment of the Fv domains of C8 and C10 to F25.S02 further shows the distinction between the binding mechanisms. When C8 is aligned with F25.S02, the two Fv domains align with an RMSD of 2.73 Å over 227 paired Cα. The C8-bound sE dimer is shifted toward the interior relative to the F25.S02-bound sE dimer (Fig. 4c,

**Fig. 2 | X-ray crystal structure of F25.S02 bound to ZikV sE. a** The X-ray crystal structure of F25.S02 Fab bound to ZikV sE dimer is shown in cartoon representation. The sE protein includes a transparent surface representation with one protomer colored to highlight domains I, II, and III and the other shown in gray. F25.S02 Fab is shown in dark green for the heavy chain and light green for the light chain. Cyan diamond represents the 2-fold symmetry axis of the dimer. **b** Surface representation of the ZikV sE dimer with the F25.S02 epitope outlined in dark green for the heavy chain and light green for the light chain. **c** Zoom in of the binding interactions of F25.S02 with domain II and the fusion loop. Hydrogen bonds are indicated by the blue dashed lines and interacting residues are labeled and shown in sticks. **d** Zoom in

of the binding interactions of F25.S02 with domain I and the N154 glycan. Hydrogen bonds are indicated by the blue dashed lines. The diagram of the N154 glycan is shown with the amount of BSA of each carbohydrate listed. Blue square represents N-acetylglucosamine, red triangle is fucose, and green circle is mannose. **e** Zoom in of the binding interactions of F25.S02 with domain III. Hydrogen bonds are indicated by the blue dashed lines. **f** BSA plot showing which residues of F25.S02 are involved in the binding to ZikV sE. The germline V(D)J genes of both heavy and light chains are shown and the CDRs are indicated. The residues involved in binding to ZikV and DenV3 sE are shown and color coded to indicate which domain they interact with.

## Table 2 | X-ray data collection and refinement statistics

| | ZikV sE bound to F25.S02 Fab (pdb_00009owf) |
|---|---|
| **Data collection** | |
| Space group | $P4_12_12$ |
| Cell dimensions | |
| *a, b, c* (Å) | 117.86, 117.86, 346.10 |
| α, β, γ (°) | 90, 90, 90 |
| Resolution (Å) | 50.00–2.30 (2.43–2.30)[a] |
| $R_{sym}$ or $R_{merge}$ | 0.227 (1.089) |
| $I / (\sigma)I$ | 15.19 (1.11) |
| $CC_{1/2}$ | 0.996 (0.482) |
| Completeness (%) | 99.5 (99.8) |
| Redundancy | 7.66 (7.08) |
| **Refinement** | |
| Resolution (Å) | 48.70–2.30 (2.32–2.30) |
| No. unique reflections | 206,169 (33,462) |
| $R_{work} / R_{free}$ | 0.225/0.274 (0.307/0.336) |
| No. atoms | 13,151 |
| Protein | 12,582 |
| Ligand/ion | 255 |
| Water | 1006 |
| *B*-factors | 68.64 |
| Protein | 68.83 |
| Ligand/ion | 84.20 |
| Water | 58.38 |
| R.m.s. deviations | |
| Bond lengths (Å) | 0.004 |
| Bond angles (°) | 0.75 |

[a]Values in parentheses are for highest-resolution shell. Data was collected from one crystal.

top). When C10 is aligned with F25.S02, the Fv domains have an RMSD of 2.50 Å over 233 paired Cα. Not only is the sE dimer shifted to the center relative to F25.S02 bound ZikV sE, but the angle of approach is rotated about 20° (Fig. 4c, bottom, Supplementary Fig. 5c). In comparison, C8 and C10 have an RMSD of 2.01 Å over 229 paired Cα, highlighting that F25.S02 is more distinct from C8 and C10 than they are from each other. The difference in binding is further shown by comparing the footprint on the ZikV sE dimer (Fig. 4d). Both C8 and C10 bind with a nearly equal split of BSA between HC and LC. C8 has a HC contribution of 46% and C10 has a HC contribution of 50% to ZikV sE, while F25.S02 is 82% HC binding by BSA (Fig. 4e). In all, F25.S02 binds a similar epitope as EDE1 antibodies but employs a distinct method of binding that is HC dependent and is confined to intra-dimer interactions.

## Discussion

Our structural analysis of F25.S02, a potent cross-neutralizing antibody against ZikV and the four DenV serotypes, reveals a common heavy chain-

dominant binding mechanism to a highly conserved patch on the E protein at the interface of the two protomers. The F25.S02 epitope encompasses domain II and the fusion loop of one protomer and domains I and III of the other protomer in the sE dimer. Interestingly, it appears that F25.S02 can readily accommodate glycans within the epitope but is not glycan-dependent for binding or neutralization. This has been observed in HIV envelope-directed antibodies where bnAbs mature to accommodate glycans but are not needed for neutralization[23–25].

The main differences in binding of F25.S02 to DenV3 and ZikV are observed at the residue insertion present in ZikV. First, the insertion in the domain I loop, which includes the glycan at position N153/N154 (DenV3/ ZikV, respectively), results in $N154_{ZikV}$ being in closer proximity to the F25.S02 Fab, with its light chain making additional glycan contacts as seen in that structure (Fig. 2d and Supplementary Fig. 3). Second, the insertion in domain III (residues 359-360 in DenV3, residues 370-373 in ZikV) also contributes to more interactions of ZikV sE with F25.S02 light chain compared to DenV3 (Fig. 2e). Additionally, EDE2 antibodies, like A11 and B7, are sensitive to glycosylation at the N153/154 location and generally have lower neutralization potentials across DenV serotypes and ZikV,[13,22] as ZikV E has a five residue insertion in the 150-glycan loop. Structural resolution of this glycan and loop have been limited to EDE2 antibodies to date[14], further demonstrating that F25.S02 is unique in the ability to accommodate the glycan while stabilizing this region among orthoflavivirus bnAbs.

The analysis suggests F25.S02 crosslinks the dimer, which would inhibit conformational changes required for viral fusion with the host membrane. This mechanism of neutralization has been reported for EDE1 bnAbs, which also cross-neutralize DenV1-4 and ZikV[14,20]. However, as we only obtained structures with the sE dimer, we cannot conclude if F25.S02 would induce a similar conformational change as EDE1 antibodies on viral particles[20]. We presently do not observe the asymmetric conformation reported before of the sE dimer and thus cannot readily explain the differences in affinity and potency observed for F25.S02 with ZikV and DenV1-4[20]. In addition, we did not obtain structures of F25.S02 with the other DenV serotypes and could not come up with a plausible explanation for the reported differences in neutralization potency across serotypes by modeling the interactions at the residue level[20].

A limitation of our study is that F25.S02 was isolated as an IgA1, unique among orthoflavivirus bnAbs, and this isotype showed a non-enhancing profile[17]. Further structural studies with the virions would need to be done to determine how F25.S02 binds to E protein as an IgA1 on the viral particle and how this increased valency of the isoform could affect binding, conformational changes or steric constraints. We nevertheless modeled F25.S02 Fab binding to the virus, by aligning our F25.S02-ZikV sE crystal structure on the whole virus cryoEM structure of ZikV (PDBid pdb_00006co8)[26] (Fig. 5a). The alignment indicated F25.S02 Fab could bind on the particle in every position on the three-dimer raft (Fig. 5c), as well as all positions around the 3-fold symmetry axis (Fig. 5b) without steric clashes. There could be a potential clash of F25.S02 light chain at the 5-fold symmetry axis (5 f position) (Fig. 5d) that could have an impact on the avidity of binding of F25.S02 antibody as seen for other EDE1 antibodies[20].

While F25.S02's epitope is similar to previously reported cross-neutralizing EDE-antibodies, its binding mode is distinct as (1) it uses predominantly heavy chain recognition, (2) its binding is shifted away from

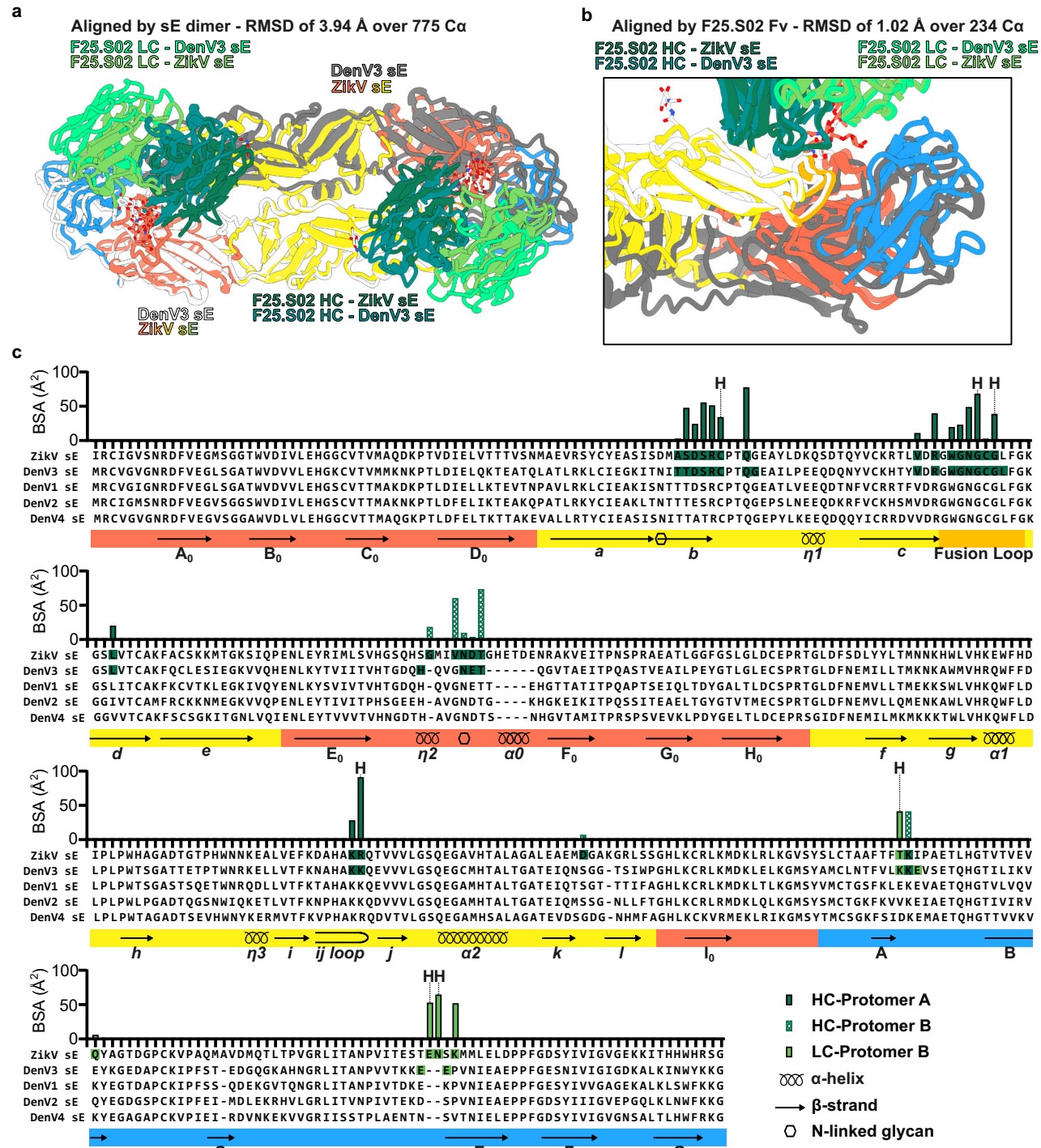

**Fig. 3 | Comparison of F25.S02 bound to DenV3 and ZikV sE. a** Alignment of ZikV and DenV3 sE structure. Alignment was made by matchmaking the sE dimer in ChimeraX. **b** Alignment of ZikV and DenV3 sE structure highlighting a single Fab-dimer interaction. Alignment was made by matchmaking the Fv domain of F25.S02 in ChimeraX. **c** BSA plot of ZikV sE bound by F25.S02 with an alignment of DenV1-4 sE. The DenV3 residues interacting with F25.S02 are highlighted in the alignment. The domains and fusion loop are colored and the secondary structural elements of ZikV sE as well as the N-linked glycan sites are shown.

the 2-fold symmetry axis of the dimer and N67 glycan in DenV3, and (3) most of its interactions are mediated by germline residues[14,20]. Thus, F25.S02 binds a similar site with a distinct mode of binding and could be a distinct class of cross-neutralizing antibody.

There is a complex interplay among B cells and antibodies of different isotypes, specificities, and functions. These interactions and their impact on pathogenesis and immunity warrant further study. Additionally, identifying other immune correlates of protection, including cell mediated[27] and non-

neutralizing antibody responses, like NS1 specific antibodies[28], are needed for improved vaccine design.

The structural information that F25.S02 binds a highly conserved epitope present on both ZikV and DenV, and that its mode of binding is primarily heavy chain dependent with little affinity maturation required for broad neutralization makes its epitope an interesting target for rational or computational vaccine design that would elicit broadly neutralizing antibodies.

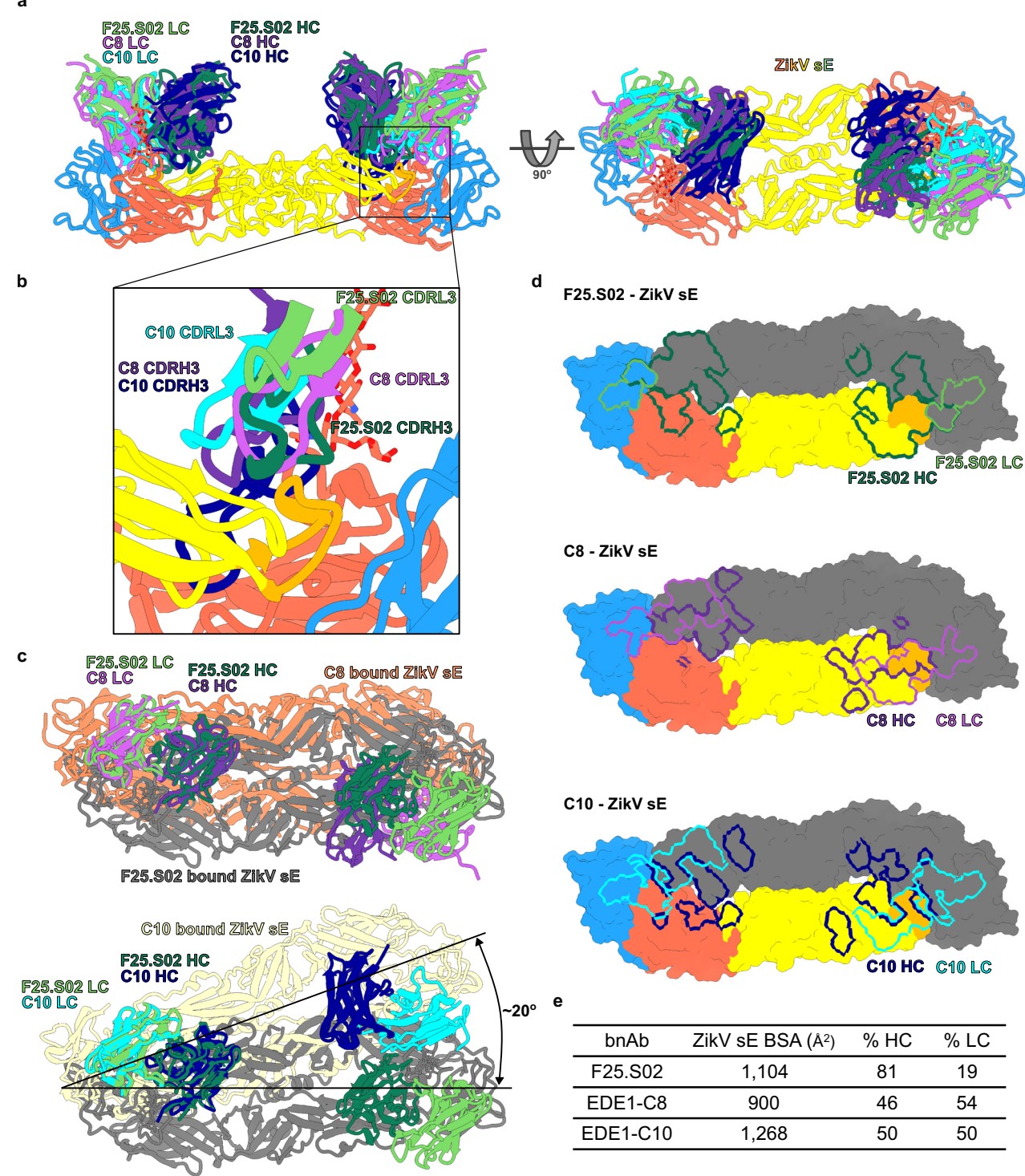

**Fig. 4 | Structural comparison of F25.S02 to other orthoflavivirus bnAb.**
**a** Alignment of EDE1 antibodies, C8 and C10, bound to ZikV sE and the F25.S02 bound ZikV sE. Alignment was made by matchmaking the sE dimer in ChimeraX. Only the F25.S02 bound ZikV sE is shown for simplicity. **b** Zoom in of the interactions of F25.S02, C8, and C10 to the fusion loop. Both C8 and C10 CDRL3 bind in the location of the F25.S02 CDRH3. **c** Alignment of C8 and C10 to F25.S02 by Fv domain of one Fab in the dimer. Alignment of C8-ZikV sE and F25.S02-ZikV sE (top). Alignment of C10-ZikV sE and F25.S02-ZikV sE (bottom). **d** Surface representation of the ZikV sE dimer with epitopes of F25.S02, C8, and C10 outlined. **e** Table showing the total BSA of each bnAb footprint on ZikV sE and the respective contribution of the heavy and light chains.

| bnAb | ZikV sE BSA (Å²) | % HC | % LC |
|---|---|---|---|
| F25.S02 | 1,104 | 81 | 19 |
| EDE1-C8 | 900 | 46 | 54 |
| EDE1-C10 | 1,268 | 50 | 50 |

## Methods
### Fab expression and purification
The gene encoding the variable region of the heavy chain of F25.S02 was cloned into a vector with a human IgG1 CH1 domain with a C-terminal his-tag using the Platinum SuperFi II DNA polymerase (ThermoFisher,

12368010) and the In-Fusion cloning system (Takara Bio, 638948). The Fab was expressed in HEK293E (RRID:CVCL_HF20) cells at $1 \times 10^6$ cells/mL using a total of 500 μg DNA per liter of culture (1:1 ratio of HC-to-LC DNA). PEI was used as the transfection reagent at a ratio of 4:1 PEI-to-DNA. Culture was harvested after 6 days and Fab was purified from cultural

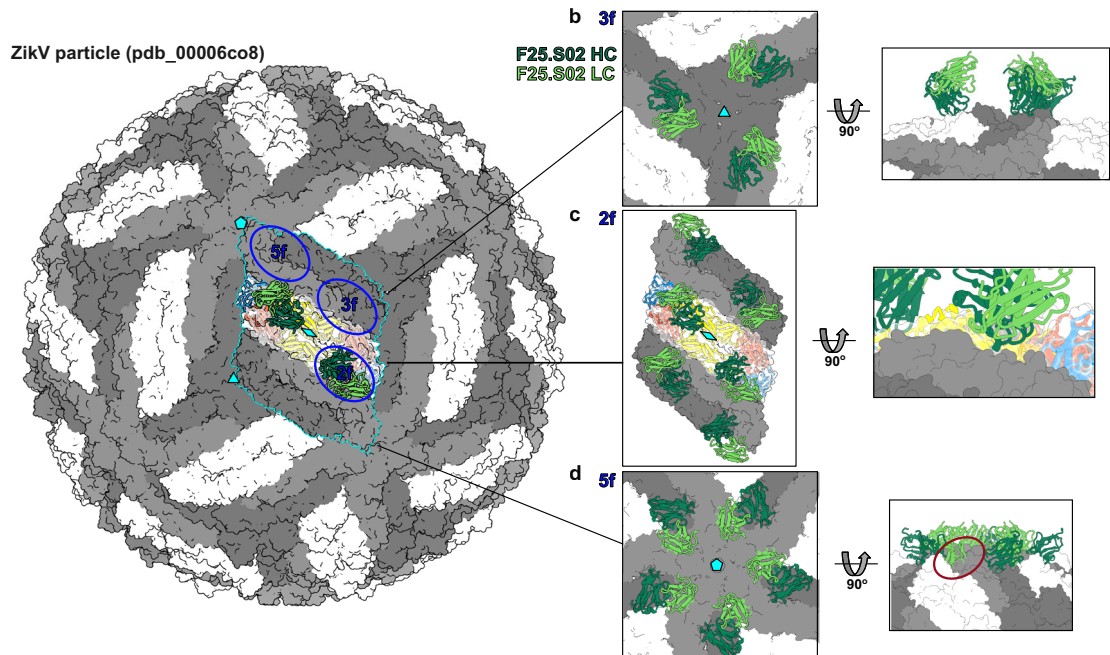

**Fig. 5 | Alignment of F25.S02 to ZikV viral particle. a** Alignment of the F25.S02-ZikV sE crystal structure on the whole virus cryoEM structure of ZikV (PDBid 6CO8). The sE raft is highlighted in cyan. The sE dimer 2-fold symmetry is shown by the diamond, the icosahedral 3-fold and 5-fold symmetry axes are shown by the triangle and pentagon, respectively. The E protein binding sites, 2f, 3f, and 5f are shown with blue circles. **b** Zoom in showing potential binding of F25.S02 at the 3f site. **c** Zoom in showing potential F25.S02 binding in an E protein raft on the particle. **d** Zoom in showing potential binding of F25.S02 at the 5f site. Light chain would potentially clash with adjacent dimer as indicated by red circle.

supernatant using His60 Ni Superflow resin (Takara Bio). Fab was eluted from column using buffer containing 50 mM Tris, pH 7.5, 300 mM NaCl, and 150 mM Imidazole. F25.S02 Fab was further purified using a HiLoad 16/600 Superdex 200 size exclusion column (Cytivia) on an AKTApure system (GE Biosciences) into a buffer containing 5 mM HEPES, 150 mM NaCl, pH 7.5.

### sE protein expression
Plasmids for WT and stabilized sE proteins were kindly provided by Brian Kulhman (University of North Carolina at Chapel Hill)[19]. DenV3 sE sc30 is based on the CH53489 reference strain E protein and has the A257C, G106D, F227W, A278P, G29K, T33V, A35M stabilizing mutations. ZikV sE was from the H/PF/2013 reference strain. sE proteins were expressed in Expi293F cells according to manufacturer's protocol (ThermoFisher, A14635). Cultures were harvested 6 days after transfection and cleared supernatant was incubated with His60 Ni Superflow resin (Takara Bio, 635660). sE protein was eluted from column using buffer containing 50 mM Tris, pH 7.5, 300 mM NaCl, and 300 mM imidazole. For structural studies, protein was further purified by size exclusion chromatography (SEC) using a HiLoad 16/600 Superdex 200 column (Cytiva) or a Superdex 200 Increase 10/300 column (Cytiva) on a AKTApure system (GE Biosciences). SEC peak corresponding to dimer was kept and concentrated using a 10 K MWCO Amicon Ultra Centrifugal Filter (Millipore Sigma, UFC901024).

The plasmid for DenV3 sE sc30 T155A was made via site-directed mutagenesis using the Platinum SuperFi II DNA polymerase (Thermo-Fisher, 12368010) and DenV3 sE sc30 T155A expressed and purified as above.

### sE + F25.S02 complex formation
For DenV3 sE sc30, sE protein was mixed with F25.S02 Fab in a 1:2.5 dimer-to-Fab ratio for 2 h at 4 °C with nutation. Complex was purified on a HiLoad 16/600 Superdex 200 column (Cytiva) and fractions corresponding to complex were pooled and concentrated to $OD_{280}$ = 8. Complex was divided into 20 µL aliquots, flash frozen in LN2, and stored at −80 °C.

For the ZikV sE WT + F25.S02 Fab complex, the plasmid for ZikV sE WT was mutated to remove the 8x His-tag using the In-Fusion cloning system (Takara Bio, 638948). Complex was expressed in Expi293F cells according to manufacturer's protocol using a plasmid DNA ratio of 2:1:1 sE-to-HC-to-LC. Culture was harvested 6 days after transfection and complex was purified from cultural supernatant using His60 Ni Superflow resin (Takara Bio). Protein was eluted using buffer containing 50 mM Tris, pH 7.5, 300 mM NaCl, and 300 mM Imidazole. Complex was further purified using a HiLoad 16/600 Superdex 200 column (Cytiva) on an AKTApure system (GE Biosciences) into a buffer containing 5 mM HEPES, 150 mM NaCl, pH 7.5. Fractions corresponding to complex were pooled and concentrated to $OD_{280}$ = 25, divided into 50 µL aliquots and flash frozen in LN2. All aliquots were stored at −80 °C.

### Biolayer interferometry
F25.S02 binding was measured using biolayer interferometry on an Octet 96 Red instrument (ForteBio) using FAB2G biosensors (Sartorius). F25.S02 Fab was diluted to 10 µg/mL in kinetics buffer (1× PBS, 0.01% Tween 20, 0.01% BSA and 0.005% NaN3, pH 7.4). Each sE protein was diluted to 1 µM in kinetics buffer. Fab was loaded onto biosensor until a threshold of 1.0 nm shift was reached. After loading, biosensors were placed in kinetics buffer for 60 s for a baseline reading. Biosensors were then immersed in analyte for 300 s in the association phase, followed by 300 s in the dissociation phase in kinetics buffer. The background signal from a biosensor loaded with Fab but with no analyte was subtracted from each loaded biosensor.

### CryoEM sample preparation
DenV3 sE sc30 + F25.S02 Fab complex was thawed and diluted to $OD_{280}$ = 5. Just prior to grid preparation, complex was mixed with 0.1% Tween20 to a final concentration of 0.25 CMC (0.5 µL of 0.1% Tween20 into 20 µL of complex). 3 µL of complex was loaded onto UltrAuFoil R1.2/1.3 300mesh grids (Ted Pella) and manually blotted. Another 3 µL was loaded onto grid and vitrified on a Vitrobot Mark IV (ThermoFisher) with a blot force of 2, blot time of 4 s at 4 °C and 100% humidity before plunging into liquid ethane.

**Article**

We found that the inclusion of detergent, Tween20 and to a lesser extent CHAPS, reduced preferred orientation. In initial screening, grids with a single blot had high concentration of particles around the edge of holes with few particles in the middle. Including a second blot before freezing forced more particles into the center of the hole. Even with these optimizations, it was still difficult to get consistent quality grids which limited the number of particles we were able to collect at high resolution.

### CryoEM data collection and processing

Data was collected on a 200 kV Glacios microscope (ThermoFisher) with a K3 direct electron detector (Gatan) at a magnification of 36,000x (1.122 Å per pixel) using SerialEM[29]. A total of 1195 movies were collected with 100 exposures over 6 s. Motion correction was performed using Warp[30] and processed using cryoSPARC (V4.5)[31]. Initial 2D templates were generated through multiple rounds of 2D classification. Using 2D templates, particles were re-extracted using template picker. A total of ~2 million particles were extracted and after 3 rounds of 2D classification, 111,111 particles were used for Ab-initio reconstruction without symmetry. The reconstruction was refined using homogenous refinement followed by non-uniform refinement[32] with C2 symmetry. Local refinement was performed using a mask to exclude the CH1/CL domain of F25.S02. The FSC curve suggested heterogeneity in the particles and a 3D classification was used to separate particles into two classes. Following homogeneous refinement, one class showed better quality and was further refined with local refinement with a mask to exclude the Fc domain and 64,110 particles. The resolution was estimated to be 4.16 Å per the gold standard $FSC_{0.143}$. Local resolution was estimated based on gold standard $FSC_{0.143}$ and is shown in Supplementary Fig. 2.

### CryoEM model building and refinement

The model of DenV3 sE protomer from PDBid 7A3P[20] was used as the initial model and a model of F25.S02 Fab was generated by AlphaFold3[33]. Both models were docked into the cryoEM map using ChimeraX (v1.9)[34]. Model building was completed using Coot (v0.9.8.1)[35] and an initial refinement was run using Phenix (v1.21) real_space_refinement[36]. Further refinement was performed using ISOLDE[37] in ChimeraX[34] and Coot[35]. Data collection and refinement statistics are summarized in Table 1. Structural figures were generated using ChimeraX and the conserved residue map was generated using the ConSurf web server[21] with the E protein sequences from the reference strains of DenV1-4 and ZikV. BSA data was determined using the PDBePISA server[38].

### Crystallization and structure determination

Initial crystallization screening was performed by sitting-drop vapor diffusion using ProPlex crystallization screen (Molecular Dimensions) and an NT8 drop setting robot (Formulatrix) at a complex concentration of $OD_{280}$ = 15. Initial crystals grew in drop F5 (0.1 M Tris, pH 8.0, 0.1 M NaCl, 8% w/v PEG 20 K). Well diffracting crystals were grown via hanging-drop vapor diffusion in 0.1 M Tris, pH 8.0, 0.1 M NaCl, 5% w/v PEG 20 K and were frozen with 40% PEG 200 as a cryoprotectant. Diffraction data was collected at Advanced Light Source beamline 5.0.3 at 12.7 keV and a temperature of 100 K. The dataset was processed using XDS[39] and AIMLESS[40] to a resolution of 2.30 Å. Matthews coefficient analysis suggested a sE dimer and two Fabs in the ASU. Initial phases were solved by molecular replacement using Phaser in Phenix[36] with a search model of ZikV sE protomer from PDBid 5JHM[41] and the AlphaFold3[33] model of F25.S02 Fab split into Fv and CH1/CL domains. Model building was completed using Coot[35] and refinement was performed in Phenix. Additional model refinement was done using ISOLDE[37] in ChimeraX[34]. The final model had Ramachandran statistic of 96.65% most favored, 2.92% allowed, and 0.43% disallowed as determined by MolProbity[42]. Data collection and refinement statistics are summarized in Table 2. BSA data was determined using the PDBePISA server[38].

### Virion modeling

To model how F25.S02 could potentially binding on the ZikV virion, we created a subset of the ZikV sE structure bound by F25.S02

consisting of the Fv and domains I and III of one protomer and domain II of the other protomer. We then aligned this substructure to each binding site in the E protein raft and around each symmetry axis of the cryoEM model of ZikV (PDBid pdb_00006co8). The alignment was made using Matchmaker in ChimeraX with default settings.

### Statistics and reproducibility

No statistical analysis was performed in the manuscript. In the case of the ZikV sE + F25.S02 crystal structure, all binding information was averaged over the two binding sites on the dimer.

### Reporting summary

Further information on research design is available in the Nature Portfolio Reporting Summary linked to this article.

### Data availability

Maps generated from the electron microscope data are deposited in the Electron Microscopy Data Bank under EMD-70931. Atomic models have been deposited in the RCSB PDB with PDB IDs pdb_00009owe and pdb_00009owf. Source Data are provided with this paper in the Supplementary Data 1 file.

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

## Acknowledgements

We thank Dr. Brian Kuhlman for providing DenV3 and ZikV sE plasmids. We thank the J. B. Pendleton Charitable Trust for its generous support of Formulatrix robotic instruments. This work was supported by a Fred Hutch Vaccine and Infectious Disease Division (VIDD) Initiative grant. Electron microscopy data were generated using the Fred Hutchinson Cancer Center Electron Microscopy Shared Resource. The EMSR is supported in part by the Cancer Center Support Grant P30 CA015704. The Berkeley Center for Structural Biology is supported in part by the Howard Hughes Medical Institute. The Advanced Light Source is a Department of Energy Office of Science User Facility under Contract No. DE-AC02-05CH11231. The ALS-ENABLE beamlines are supported in part by the National Institutes of Health, National Institute of General Medical Sciences, grant P30 GM124169. A portion of this research was supported by the NIH S10 instrumentation grant 1S10OD028581-01.

## Author contributions

N.K.H., J.L., L.G., and M.P. conceived the project. N.K.H., J.L., L.G., and M.P. designed the experiments. N.K.H. performed the cloning, expressed and purified the proteins, formed the complexes, collected and processed the cryoEM data, and built and refined the model. N.K.H. crystallized the protein, collected and processed the diffraction data, and solved the crystal structure. N.K.H., J.L., L.G., and M.P. analyzed and discussed data. N.K.H. and M.P. wrote the original manuscript draft. N.K.H., J.L., L.G., and M.P. reviewed and edited the manuscript. N.K.H., J.L., L.G., and M.P. secured funding for the project.

## Competing interests

J.L. and L.G. are inventors on a patent application filed by Fred Hutchinson Cancer Center relating to F25.S02. L.G is an employee and a possible shareholder of Vaccine Company, Inc. The other authors have no competing interests.
