## [Transparent Peer Review file · Communications Biology]

Structural basis for antibody cross-neutralization of dengue and Zika viruses

Corresponding Author: Dr Marie Pancera

Version 0:

Reviewer comments:

Reviewer #1

(Remarks to the Author)

Summary

This manuscript provides high-quality cryo-EM and X-ray structural data revealing how the mAb F25.S02 cross-neutralizes all four dengue virus serotypes and Zika virus. The authors convincingly show that F25.S02 targets a conserved E-dimer epitope via a heavy-chain-dominant binding mode distinct from known EDE1 antibodies. The study has clear implications for structure-guided vaccine design and antibody engineering in Orthoflavivirus research. The work is technically rigorous and of substantial interest to the flavivirus field and vaccine development.

1. RMSD analysis and structural alignment (Fig. 3; lines ~149–153)

The manuscript reports 3.94 Å RMSD after aligning the entire sE dimer (775 C α) and 1.02 Å RMSD after aligning the F25.S02 Fv (234 C α). Both are global comparisons. Because sE dimers adopt multiple conformations due to DI–DII hinge flexibility (further influenced by crystal packing), global alignments can overstate differences that are not directly related to epitope recognition or antibody engagement. The authors already acknowledge this variation: “DenV3-bound F25.S02 Fv is shifted... mostly due to slight differences in sE conformations.”

To better highlight functionally relevant structural differences, please complement the global RMSDs with a localized alignment restricted to residues within ~5–8 Å of the Fv interface. This would capture local epitope variation between DENV3 and ZIKV while minimizing the influence of hinge movements and packing effects. In addition, aligning on a single epitope-side of the dimer (i.e., DI/III from protomer A plus DII from the partner protomer B) could reveal subtle differences in Fv engagement geometry between the two complexes and provide a more accurate measure of similarity and difference.

2. Apply the same principle to Fig. 4 (C8/C10) Please re-evaluate approach angles after aligning a single E-dimer epitope-side. Angle differences calculated across both epitopes in the dimeric sE context can be confounded by natural sE hinge motion and packing artifacts, and are not representative of the virion E conformation (in Fig. 4c and related text). Aligning to one epitope-side would better distinguish genuine differences in approach angle and Fv position on the dimer.

3. Fig. 5 modeling note Since the virion E protein differs conformationally from isolated sE, please align the virion E epitope to the corresponding epitope-side from the ZIKV crystal structure before modeling Fab placement. Also please specify the alignment method and residue ranges used in the Methods section.

4. Asymmetric unit composition and intra-ASU asymmetry The ZIKV sE–F25.S02 crystal contains an sE dimer and two Fabs in the asymmetric unit (ASU), and the text notes no asymmetric conformation (Discussion, lines 228–230). To support this statement, please briefly describe how asymmetry was assessed—for example, by aligning the ZIKV Fv bound at site 1 with the Fv bound at site 2 and reporting an epitope-specific RMSD between the two binding sites. A small numerical difference would clearly support the observation that both Fabs bind equivalently within the ASU.

5. Place findings in context of EDE2 antibodies In the Discussion (lines ~205–208), the manuscript highlights that F25.S02 tolerates the ZIKV-specific insertion and glycan positioning. It would be valuable to add that EDE2 antibodies are limited in their breadth precisely because this insertion repositions the glycan and prevents ZIKV recognition. Emphasizing that F25.S02 overcomes this structural constraint—unlike EDE2 bnAbs—would strengthen the discussion and underscore its mechanistic distinctiveness and potential importance for pan-flavivirus immunogen design.

6. DENVs and ZIKV Strain and sequence information For reproducibility and to help readers relate structures to neutralization/binding data, please specify the exact DENV3 and ZIKV strains used to express the sE constructs (strain names and/or GenBank/UniProt accession IDs). Likewise, list the reference strains or accession sets used for the ConSurf conservation analysis.

7. In the cryo-EM Methods and related supplemental figure captions, “Fc domain” is mentioned in the context of masking

during refinement. Because the analyzed construct is a Fab, the masked constant region corresponds to CH1/CL rather than Fc (CH2/CH3). Please update the terminology to "CH1/CL domains" for accuracy and to maintain consistency with the X-ray methods section.

This is a careful and timely structural study with clear implications for vaccine design. The suggested clarifications aim to refine interpretation and reproducibility and do not require new experiments.

Version 1:

Reviewer comments:

Reviewer #1

(Remarks to the Author)

Thank you for the revised manuscript. The work is clearly presented, and the revisions are well executed. I have no further comments and am pleased to recommend the manuscript for publication.

REVIEWER COMMENTS

Reviewer #1 (Remarks to the Author):

The authors present structural characterization of F25.S02 antibody against dengue (1-4) and Zika viruses. F25.S02 is a broadly neutralizing, E protein reacting antibody, previously isolated by the same authors. Previous work using immunoassays and scanning mutagenesis of VLPs suggested that F25.S02 recognizes E-dimer tertiary epitope. Current work utilizes X-ray crystallography and Cryo-EM, single particle analysis of immune complexes of monovalent fragments of F25.S02 with dengue serotype 3 soluble E stabilized-dimer and with wtZika virus soluble E dimer. The authors further support structural analysis using size exclusion chromatography of immune complexes and Biolayer Interferometry (BLI) binding assays.

The authors present thorough analysis of obtained structures and compare mode of interaction to previously characterized EDE1 C10 and C8 bnAb antibodies. Furthermore, in their comparative analysis the authors compare atomic details of F25.S02 binding to Zika soluble E dimer with previously published structure SIgN-3C antibody with Zika virions, whereby SIgN-3C recognizes tertiary epitope different from E dimer epitope – but rather as previously shown inter dimer complex epitope.

The authors show that F25.S02 binds E dimer epitope (EDE) similar to EDE1, yet with specific differences that are described in details in the manuscript. The F25.S02 targeted epitope falls into the definition of EDE epitope and quite significantly overlaps with EDE1 and EDE2, yet it has following distinctive features: (1) majority of amino acid contacts on E dimer interface are contributed by variable heavy chain, (2) it is positioned a bit laterally than EDE1 and EDE2, even though majority of the footprint overlaps (3) the structure of immune complexes shows resolved contacts with N153 glycan, while EDE1 bnAbs were previously shown to displace this glycan and it was not resolved in the published EDE1 complexes with dengue and Zika E soluble dimers and virions. The type of associations with N153 glycan, described for F25.S02 is more resembling those described for EDE2 bnAbs, yet previous work distinguished F25.S02 from EDE2, because its neutralization was not abrupt upon mutating N153 glycosylation site, as opposed from EDE2 bnAbs. These features, as suggested by the authors distinguish F25.S02 from EDE1 and EDE2 bnAbs and define it as another EDE bnAb, yet with strong epitopic overlap with EDE1 and EDE2.

The work is solid and novel, presenting the structural analysis of the mechanism of interaction of a broadly neutralizing Ab (anti-dengue serotypes (1-4) and anti-Zika) with extremely potent neutralizing activity.

The work, as such merits publication in Nature Communications after addressing the following experimental and textual points.

We thank the reviewer for this nice summary of our work

Major points:

1) Supplementary Figure 1:
Quantitative analysis of the binding/dissociation curves of BLI experiments is missing (KD/Kon/Koff) for

the tested Fab fragments and mutants of sE. Binding data is for wt dengue serotype 3 (in addition to the stabilized dimer), either by SEC, or by BLI would be helpful. SEC shifts do not directly indicate stoichiometry. Multiple Angle Light scattering would significantly strengthen the presented results. PAGE analysis of the shifted peak in SEC should be presented.

We agree that a more thorough BLI/kinetics analysis with the WT DenV3 sE protein would be good but due to expression issues it is not feasible to get the replicates and concentration breadth needed for high confidence measurements. The previous work (ref 17 and 22) has already done neutralization experiments which showed similar or better IC_{50} values as other EDE binding bnAbs. Neutralization experiments are more immunologically relevant as binding affinity does not necessarily correlate to protection in vivo.

While we have included SDS-PAGE analysis of both complexes in Fig S1, we do not have access to MALS and feel it is not necessary to support our main conclusions as the SEC shows adequate shift and both structures clearly shows the SE dimer with two Fabs bound.

2) The paper would strongly benefit from mutagenesis analysis of F25.S02 variable loops on the antibody side, or alternatively, including binding or neutralization data on natural antibody variants (clonal relatives) of F25.S02, identified by sequencing efforts in the earlier papers by the authors. If such data on binding or neutralization is presented for dengue serotypes and Zika, the authors would make conclusions related to the significance of contacts made by antibody residues observed by the structures to actual binding or neutralization. Mutagenesis analysis of antibody either to assess contribution of specific residues or in an attempt to generate super binders, would strengthen the physiological and biochemical aspects of the current work.

While this could lead to some interesting results, we feel that this is outside of the scope of this manuscript which was to determine how F25.S02 binds to DenV3 sE and ZikV sE and how it compares to other bnAbs. These additional experiments would also be an undue financial and time burden.

3) Table S2:

For Zika sE F25.S02 Fab complex, Rwork/Rfree statistics seems to be high for this resolution.

We agree that the Rwork/Rfree are higher than desired for this resolution. There are several blobs in the Fo-Fc density that were left unmodeled and we believe this is resulting in the inflated refinement statistics. After extensive trial and error to fit this density with crystallization solution components, cryoprotectant or even molecules that could have been carried over from expression media, we decided to leave the density empty rather than over fit or misidentify.

4) No data is presented in the current paper for non-stabilized Dengue serotype 3, without N153 glycan. No such data is presented for Zika sE mutant in N154 either. Including these data in the manuscript would be helpful to decipher the actual contribution of the 150 loop glycan, in particular since the analysis of the atomic contacts presented by the authors suggests that this glycan contributes: ' ~213 Å² of buried surface area (18% of the total BSA for DenV3)'.

We agree that this data would be good to include but exceedingly difficult with the non-stabilized sE dimers. The non-stabilized dimers express at incredibly low yields and fall apart to mostly monomers

rapidly. The previous works (ref 17 and 22) showed increased neutralization potential of F25.S02 against reporter viruses lacking the glycan showing that the glycans are not required for neutralization and suggest that F25.S02 can bind in the presence of the glycan (since it shows contacts as shown by the BSA) but does not require the glycan.

5) In general, the paper would benefit from including a systemic analysis of different regions of sE and antibody variable loops by their contribution in BSA (surface and percentage), summarized in a table, presented in supplementary. Including such table would assist systematic presentation and analyses and would support the data presented in plot of Figure 3 on the sequence alignment.

We thank the reviewer for this suggestion and have made a supplementary table (Supplementary Table 3) summarized as pie charts in Supplementary Figure 3.

6) It would be important to include in the manuscript a table that would summarize interactions presented by antibody loops (e.g. contact map or contact table, including list of salt bridges, hydrogen bonds, etc). This would help a reader to follow the graphical representation of contacts in the main figures and analyze the important contacts.

We have created a table and included it in the supplemental, Table S4.

7) The comparison to SigN-3C seems to be not necessarily informative. Since this antibody is known to recognize inter-dimer epitope. Furthermore, because the current work does not present any analyses of cryoEM of F25.S02 with virion particles, such comparison may be misleading. I would suggest not including this comparison in the paper, if no additional data on cryoEM of F25.S02 complexed with dengue or Zika virion particles is added.

We have removed this section.

Minor points:

1) Please avoid using green colors and different shades of it in structural figures due to difficulties for color blinded (e.g. Figures 1-5).

We took color blind accessibility into consideration when making the figures and tried to pick colors that are separate enough to be easily viewable. We have run all figures through a colorblind image tester (https://bioapps.byu.edu/colorblind_image_tester) that tests the friendliness of images to people with moderate-to-severe, red-green colorblindness. All figures passed with a friendly grade and a confidence over 95%. If the reviewer or editor has concerns over the coloring, we can modify the figures.

2) In Figure 2, Arg 99 seems not to be colored by atoms, as other labeled residues do, this is confusing, please modify.

We thank the reviewer for catching the confusing aspect of the figure. Arg99 is colored by atoms but the label is near Val97. We have added arrows to clarify.

3) Please improve textual labels in structural figures because they do not pop up, possibly make thicker black outline or change the color of yellow to be darker.

We have modified figures to increase the outline thickness and darkened the yellow color of domain III.

4) Page 9, lines 195-197, sentence:

‘However, as we only obtained structures with the sE dimer, we cannot conclude if F25.S02 would induce a similar conformational change as EDE1 antibodies on viral particles [17].’

It is not clear what is the relevance of reference [17], seems that it is cited here by mistake?

We thank the reviewer and have updated the reference to the correct one [20].

Reviewer #2 (Remarks to the Author):

This work elucidates the molecular mechanism by which the broadly neutralizing antibody (bnAb) F25.S02 neutralizes dengue virus (DENV) serotypes and Zika virus (ZIKV). Using cryo-EM of a stabilized DENV3 sE dimer and X-ray crystallography of the ZIKV sE protein, the authors provide two structural snapshots that add to the still-limited catalogue of DENV/ZIKV antigen-bnAb complexes. The work also delivers the first cryo-EM structure of a sE dimer / Fab complex, demonstrating a rapid, biosafe workflow. The study is timely and valuable; however, deeper mechanistic insight will require additional experiments and tighter integration with earlier findings if the work is to inform structure-guided vaccine design.

We thank the reviewer.

1. The manuscript repeatedly states that “a safe and effective vaccine must elicit antibodies that simultaneously neutralize DENVs and ZIKV.” This is an oversimplification. Balanced, serotype-specific neutralizing antibody responses to each of the four DENV serotypes can be protective, and other immune components—T-cell responses and anti-NS1 antibodies, for example—also contribute to immunity. Moreover, some cross-reactive bnAbs (including EDE2 A11 and EDE1 B2 in their IgG form) can enhance infection. Please acknowledge this complexity and clarify that eliciting bnAbs is only one strategy for achieving balanced, non-enhancing protection.

We chose to emphasize neutralizing antibodies both to highlight the fact that they are the focus of most DENV vaccines in advanced clinical development and to provide appropriate context for the goal of our study, which was to characterize the structural epitopes of broadly neutralizing antibodies. Importantly, the presence of bnAbs targeting the E-dimer epitope have been shown to reduce risk of symptomatic dengue in humans (DOI: 10.1126/scitranslmed.adq0571). Nevertheless, we agree that protective immunity against DENV will most certainly involve additional immunological responses and have modified the discussion (lines 218-221) as follows: “There is a complex interplay among B cells and antibodies of different isotypes, specificities, and functions. These interactions and their impact on pathogenesis and immunity warrant further study. Additionally, identifying other immune correlates of protection, including cell mediated [27] and non-neutralizing antibody responses, like NS1 specific antibodies [28], are needed for improved vaccine design.”

2. F25.S02 binds the prefusion E-dimer epitope (EDE) originally defined by Dejnirattisai et al. and structurally characterized by Rouvinski et al.—an EDE1 region that does not depend on the N-linked glycan at the 150-loop for binding or neutralization. The text should explicitly assign F25.S02 to the EDE1 class. Dejnirattisai et al. reported several EDE1 mAbs from diverse germline lineages, which are naturally expected to adopt distinct engagement angles and contact footprints while targeting this shared epitope.

We thank the reviewer for this suggestion however one of the major points of this manuscript is that F25.S02 is distinct from these EDE1 antibodies, and we believe that the Dejnirattisai et al definition of EDE1 is too broad. We think that F25.S02 represents a distinct class of EDE binding antibodies that is vastly heavy chain dependent and requires lower rates of somatic hypermutation, something that has not been shown for EDE1 class antibodies, and as such distinguishes this antibody from the others. There is a precedence for this type of distinction in the HIV field with different classes of CD4 binding site targeting antibodies.

3. The structure shows extensive contacts with the 150-loop glycan, yet neutralization improves when the glycan is absent (ref. 22), suggesting these interactions are opportunistic rather than essential. Binding-affinity measurements (e.g., BLI) on glycosylated versus deglycosylated sE dimers would clarify the glycan's contribution. When comparing F25.S02 with EDE1 antibodies C8 and C10, note that the 150-loop remains in a “capping” conformation in the F25.S02 complex, in contrast to the displaced loop in C8/C10; this difference could underlie their divergent glycan dependence and should be discussed.

As the reviewer has pointed out, F25.S02 has already been shown to be a better neutralizer in the absence of the 150-loop glycan. While binding kinetics data could show differences in K_{on}/K_{off} , it would be limited to the stabilized versions of the sE proteins and we do not see how this would clarify the glycan's contribution in a way that the neutralization data has not already shown.

We have added more discussion of the binding to the 150-loop (lines 160-163): “In the structures for C8 and C10, the 150-glycan loop is displaced by the CDRH3, and the glycan is not resolved. While in the F25.S02 structures, the glycan is well-resolved and the CDRH1 interacts with the 150-loop staying in a capping position and the CDRH3 shifting to the top of the fusion loop relative to C8 and C10 further suggesting a different mode of binding.”

4. Supplementary Figure 1 reports binding at a single antigen concentration (1 μ M). In the absence of structures for other serotypes, full binding curves for all four DENV serotypes and for ZIKV are needed to correlate affinity with reported neutralization potencies and to assess possible avidity effects and neutralization differences—for example, for DENV2, which is neutralized less efficiently in both laboratory and clinical strains.

As mentioned in response to reviewer 1, full kinetics work ups could be difficult due to expression yields and the data would be limited to stabilized sE dimers. We also believe that kinetics of the stabilized sE dimers is not necessarily useful here as kinetics can vary wildly with minor changes to the protein. Ideally, we would have done kinetics with the WT sE dimers but this is not possible due to low expression yields and quick breakdown to monomers.

5. F25.S02 was isolated as an IgA1, making it unique among reported flavivirus bnAbs, which are typically IgG. This isotype confers a non-enhancing profile. The authors should discuss how mechanistic insights from a monovalent Fab snapshot translate to the full-length IgA1 context, considering its higher-order valency (dimeric or secretory forms), hinge flexibility, and potential steric constraints when engaging virion. Please indicate whether the Fab used for the structural studies carries an IgG1- or IgA1-derived CH1 domain.

We thank the reviewer for bringing this up but do not think we can address this with the current data as differences between IgA and IgG would likely only be resolved at the virion level. The F25.S02 Fab in the structures had a IgG1-derived CH1 domain and we have clarified this in the methods.

We have noted this limitation in the discussion section (lines 202-205): “A limitation of our study is that F25.S02 was isolated as an IgA1, unique among orthoflavivirus bnAbs, and this isotype showed a non-enhancing profile [17]. Further structural studies with the virions would need to be done to determine how F25.S02 binds to E protein as an IgA1 on the viral particle and how this increased valency of the isoform could affect binding, conformational changes or steric constraints.”

6. Figure 2f should list complete V(D)J gene assignments, not just the V gene. To support the claim that germline-encoded residues dominate the interface, the authors should include binding and/or neutralization data for the inferred germline variant.

We have extended the sequence to include the complete inferred V(D)J assignments in Fig.2. While binding or neutralization data of germline reverted F25.S02 could be useful in assessing the effect of maturation on breadth, we feel that it is outside the scope of this manuscript and would be an undue hardship as we no longer have the ability to test neutralization at our institute.

7. This appears to be the first cryo-EM structure of a flaviviral soluble-E dimer, providing a rapid, crystallization-free method that avoids the need for inactivated virions or BSL-2/3 facilities. The Methods section should briefly outline key technical details—grid preparation (e.g., whether detergent was required and whether it mitigated preferred orientation), the main factors limiting resolution (such as conformational heterogeneity), and practical tips for improving resolution—to help other groups replicate the approach.

We have expanded on our methods section for grid preparation.

8. Both structures present substantial validation issues, explicit hydrogens at moderate-to-low resolutions are problematic and unjustifiable. The discrepancy in EM resolution estimation (reported vs. recalculated) requires attention. The resolution reported (4.16 Å) may be overly optimistic. Carbohydrate structures require geometry refinement.

The addition of hydrogens is necessary for refinement using ISOLDE. We can remove them but feel that inclusions of hydrogens have become standard in the cryoEM field regardless of resolution. Binding analysis programs like PDBePISA ignore hydrogens so interacting residue identification was performed with hydrogens excluded. The difference in the resolution estimation is due to the recalculated FSC using the unsharpened half maps but the reported is using a sharpened mask. We are unclear what is meant by

the carbohydrates needing geometry refinement. In both structures, the carbohydrates fit in the density well with no chiral outliers and very little torsion or bond length outliers.

Reviewers' comments:

Reviewer #1 (Remarks to the Author):

Summary

This manuscript provides high-quality cryo-EM and X-ray structural data revealing how the mAb F25.S02 cross-neutralizes all four dengue virus serotypes and Zika virus. The authors convincingly show that F25.S02 targets a conserved E-dimer epitope via a heavy-chain-dominant binding mode distinct from known EDE1 antibodies. The study has clear implications for structure-guided vaccine design and antibody engineering in Orthoflavivirus research. The work is technically rigorous and of substantial interest to the flavivirus field and vaccine development.

1. RMSD analysis and structural alignment (Fig. 3; lines ~149–153)◇

The manuscript reports 3.94 Å RMSD after aligning the entire sE dimer (775 Cα) and 1.02 Å RMSD after aligning the F25.S02 Fv (234 Cα). Both are global comparisons. Because sE dimers adopt multiple conformations due to DI–DII hinge flexibility (further influenced by crystal packing), global alignments can overstate differences that are not directly related to epitope recognition or antibody engagement. The authors already acknowledge this variation: “DenV3-bound F25.S02 Fv is shifted... mostly due to slight differences in sE conformations.”

To better highlight functionally relevant structural differences, please complement the global RMSDs with a localized alignment restricted to residues within ~5–8 Å of the Fv interface. This would capture local epitope variation between DENV3 and ZIKV while minimizing the influence of hinge movements and packing effects. In addition, aligning on a single epitope-side of the dimer (i.e., DI/III from protomer A plus DII from the partner protomer B) could reveal subtle differences in Fv engagement geometry between the two complexes and provide a more accurate measure of similarity and difference.

We thank the reviewer for this suggestion. Aligning residues on sE within 5Å of Fv has 0.99 Å RMSD over 36 Cα. Aligning DI/III of protomer A plus DII of protomer B has 1.36 Å RMSD over 320 Cα. We have added language to include this information on line 142 “Aligning domains I and III of protomer A and domain II of protomer B of DenV3 sE to the same regions of ZikV sE shows a much more similar structure with an RMSD of 1.36 Å over 320 paired Cα. An even more localized alignment using all residues of the sE proteins within 5Å of F25.S02 had an RMSD of 0.99 Å over 36 Cα.”

2. Apply the same principle to Fig. 4 (C8/C10)◇

Please re-evaluate approach angles after aligning a single E-dimer epitope-side. Angle differences calculated across both epitopes in the dimeric sE context can be confounded by natural sE hinge motion and packing artifacts, and are not representative of the virion E conformation (in Fig. 4c and related text). Aligning to one epitope-side would better distinguish genuine differences in approach angle and Fv position on the dimer.

In the alignments with C8 and C10, panel A shows the alignment of a single epitope on sE and it is harder to see the differences in angle of approach. We chose to show the alignment of the Fv domains as it was clearer to see the difference in the angle of approach. The ~20° value difference is an average of aligning

both epitopes of C10 to F25.S02. We have made an additional supplemental figure (Figure S5) where we used the alignment from Fig4a and used ChimeraX to generate a plane through the center of each Fv using. Using this method, we were able to calculate the difference in angle between the plans and the average of the two sites of C10 and F25.S02 was 24.6°.

3. Fig. 5 modeling note◇ Since the virion E protein differs conformationally from isolated sE, please align the virion E epitope to the corresponding epitope-side from the ZIKV crystal structure before modeling Fab placement. Also please specify the alignment method and residue ranges used in the Methods section.

We modeled the binding on the virion similarly to how the reviewer suggested we perform alignments in comment 1. We generated a subset of the structure that was DI and DIII of protomer A and DII of protomer B and the Fv. We then used Matchmaker in ChimeraX with default settings to align the structure to each binding site on a E protein raft or symmetry axis. We have added a method section for these alignments.

4. Asymmetric unit composition and intra-ASU asymmetry◇ The ZIKV sE–F25.S02 crystal contains an sE dimer and two Fabs in the asymmetric unit (ASU), and the text notes no asymmetric conformation (Discussion, lines 228–230). To support this statement, please briefly describe how asymmetry was assessed—for example, by aligning the ZIKV Fv bound at site 1 with the Fv bound at site 2 and reporting an epitope-specific RMSD between the two binding sites. A small numerical difference would clearly support the observation that both Fabs bind equivalently within the ASU.

ASU composition was evaluated by Matthews coefficient analysis when looking at diffraction data quality. The slight asymmetry difference in the binding are highlighted in tables S3 and S4 and show differences in the overall BSA of each Fab binding and slight differences in bond length. We have added a sentence to the results to specify this “When aligning the two Fabs the RMSD is 1.03 Å over 394 Cα with the biggest differences coming from the CH1/CL domain and when aligning just the residues involved in binding the RMSD is 0.218 Å over 23 Cα, indicating that the binding sites are in good agreement.” (line 112)

5. Place findings in context of EDE2 antibodies◇ In the Discussion (lines ~205–208), the manuscript highlights that F25.S02 tolerates the ZIKV-specific insertion and glycan positioning. It would be valuable to add that EDE2 antibodies are limited in their breadth precisely because this insertion repositions the glycan and prevents ZIKV recognition. Emphasizing that F25.S02 overcomes this structural constraint—unlike EDE2 bnAbs—would strengthen the discussion and underscore its mechanistic distinctiveness and potential importance for pan-flavivirus immunogen design.

We initially chose to leave out discussion of EDE2 bnAbs in this manuscript as they are glycan dependent and F25.S02 is more closely related to the EDE1 bnAbs. We have added to the discussion to include a reference to EDE2 bnAbs “Additionally, EDE2 antibodies, like A11 and B7, are sensitive to glycosylation at the N153/154 location and generally have lower neutralization potentials across DenV serotypes and ZikV. EDE2 antibodies have been the only ones to structurally resolve this loop and glycan to date,

further demonstrating that F25.S02 is unique in the ability to accommodate and stabilize this region among orthoflavivirus bnAbs.”

6. DENVs and ZIKV Strain and sequence information◊ For reproducibility and to help readers relate structures to neutralization/binding data, please specify the exact DENV3 and ZIKV strains used to express the sE constructs (strain names and/or GenBank/UniProt accession IDs). Likewise, list the reference strains or accession sets used for the ConSurf conservation analysis.

We have updated the methods section for sE protein expressions to include the strain names for DenV3 and ZikV. DenV3 sE sc30 is based on the CH53489 reference strain E protein and has the A257C, G106D, F227W, A278P, G29K, T33V, A35M stabilizing mutations. ZikV sE was from the H/PF/2013 reference strain. We have added the strains used for the ConSurf analysis on line 85: “using E protein sequences of reference strains of each DenV serotype and ZikV (DenV1: WP-74, DenV2: 16681, DenV3: CH53489, DenV4: TVP376, ZikV: H/PF/2013).”

7. In the cryo-EM Methods and related supplemental figure captions, “Fc domain” is mentioned in the context of masking during refinement. Because the analyzed construct is a Fab, the masked constant region corresponds to CH1/CL rather than Fc (CH2/CH3). Please update the terminology to “CH1/CL domains” for accuracy and to maintain consistency with the X-ray methods section.

We have changed Fc domain to CH1/CL domain and updated the cryoEM processing figure with the same change.

This is a careful and timely structural study with clear implications for vaccine design. The suggested clarifications aim to refine interpretation and reproducibility and do not require new experiments.